# Environmental Salinity Affects Growth and Metabolism in Fingerling Meagre (*Argyrosomus Regius*)

**Ignacio Ruiz-Jarabo** [1,*,†], **Ana Belén Tinoco** [2,†], **Luis Vargas-Chacoff** [3],
**Juan Antonio Martos-Sitcha** [1], **Ana Rodríguez-Rúa** [2], **Salvador Cárdenas** [2] **and**
**Juan Miguel Mancera** [1]

[1] Department of Biology, Faculty of Marine and Environmental Sciences, Instituto Universitario de Investigación Marina (INMAR), Campus de Excelencia Internacional del Mar (CEI-MAR) University of Cadiz, 11510 Puerto Real, Cádiz, Spain; juanantonio.sitcha@uca.es (J.A.M.-S.); juanmiguel.mancera@uca.es (J.M.M.)

[2] Departamento de Producción, IFAPA Centro El Toruño, Junta de Andalucía, E-11500 El Puerto de Santa María, Cádiz, Spain; anab_tinoco@yahoo.es (A.B.T.); anarodriguezrua@juntadeandalucia.es (A.R.-R.); pagurta@gmail.com (S.C.)

[3] Instituto de Ciencias Marinas y Limnológicas, Facultad de Ciencias, Universidad Austral de Chile, Casilla 567, Valdivia, Chile; luis.vargas@uach.cl

\* Correspondence: ignacio.ruizjarabo@uca.es; Tel.: +34-6155-14755

† These authors contributed equally to this work.

**Abstract:** The meagre (*Argyrosomus regius*), a farmed fish in Mediterranean countries, seasonally migrates from offshore areas to estuaries for reproduction. During the first two years of life, the meagre evidences a certain grade of euryhalinity by staying in brackish waters close to the shore. The aim of the present study was to establish if fingerling growth in brackish water is improved compared to seawater, where current culture procedures are conducted. Three-month-old fingerlings were maintained for 45 days under two different salinity regimens (12 and 39 ppt). Several growth parameters as well as osmoregulatory and metabolic variables were assessed. Specific growth rate and hepatosomatic index values revealed that fingerlings performed better in brackish waters (12 ppt) compared to 39 ppt. This study contributes to optimizing meagre rearing conditions, thereby supporting the role of *A. regius* in aquaculture diversification.

**Keywords:** *Argyrosomus regius*; fingerling; growth; meagre; osmoregulation; salinity

---

## 1. Introduction

The meagre (*Argyrosomus regius* Asso, 1801) is a teleost fish within the Sciaenidae family. It is a good candidate for aquaculture diversification in Mediterranean countries [1,2] due to its fast growth rates [3] and lean flesh [4]. However, in contrast to other Sciaenidae species that are widely farmed in countries such as China [5], meagre culture evidenced variable growth rates [6] and its production is still quite limited (Food and Agriculture Organization of the United Nations, 2015).

This species, just as other sciaenids [7], seasonally gathers in estuaries and coastal lagoons to spawn [8,9]. Moreover, juveniles of *A. regius* [10], *A. hololepidotus* [11], *A. japonicus* [12], and other sciaenids [13] inhabit brackish waters during the initial lifecycle stages, with adults migrating to marine habitats [12]. Therefore, river water flows of freshwater to estuaries are vital for the survival of these Sciaenidae species [14]. Importantly, various circumstances, such as dam construction,

heavy rains, or droughts, can either increase or decrease the salinity of estuaries, thereby affecting fish osmoregulation and, consequently, recruitment.

Osmotic costs in teleosts are highly variable, but between 10% and 50% of energy expenditure in teleosts is devoted to osmoregulation [15]. It has been demonstrated in several species that environmental salinities close to the iso-osmotic point minimized energy demand for osmoregulation, thus enhancing fish growth [16–18]. In juvenile *A. regius*, previous studies suggested that salinities closer to the iso-osmotic point in this species (12 ppt) lowered osmoregulatory work [19] and favor somatic growth by increasing insulin-like growth factor 1 (*IGF1*) mRNA expression [20]. This may explain why juveniles of some *Argyrosomus* species inhabit estuaries and brackish coastal waters during the first years of life [12,21]. However, *A. japonicus* juveniles weighing less than 10 g perform best at a salinity of 35 ppt [22], while 10–20-day-old larvae show optimum growth at 5–12.5 ppt [23]. This information supports previous studies findings that the salinity for optimum growth can vary with fish age [24]. While some information exists for the culture of *A. regius* juvenile fish in cages and earthen-ponds with natural seawater conditions [3,25–27], there is less information available for the effects on *A. regius* production at different salinities during the early lifecycle stages [19].

To acclimate to variations in osmotic condition, fish undergo an adaptive period during which allostatic changes occur. These processes, mediated by glucocorticoid hormones such as cortisol [28], include modifications in the osmoregulatory system and the reallocation of energy resources [29]. After this period, a chronic regulatory period is achieved and osmoregulatory and metabolic parameters reach homeostasis [30]. Gills, kidneys, and the digestive tract are the most energetically demanding tissues in terms of osmoregulation [15], mostly due to the activity of the $Na^+/K^+$-ATPase enzyme in maintaining plasma $Na^+$ and $Cl^-$ levels within a narrow range [31]. In fish, the liver supports the energy expenditures of osmoregulatory tissues by metabolizing carbohydrates and lipids [32,33]. In these tissues, glucose is the main energy source but other metabolic substrates can be also used to cope with stressing environmental conditions [34].

Altogether, information regarding the culture of *A. regius* fingerlings is scarce and limited to seawater, with no information available about their culture in iso-osmotic environments, where natural growth occurs in wild fish. Thus, the aim of this study was to establish if environmental salinities close to the iso-osmotic point for this species, results in better growth rates for *A. regius* fingerlings.

## 2. Results

Weight gain of *A. regius* fingerlings maintained at 12 and 39 ppt salinity for 45 days is shown in Figure 1A. After 17 days, the 12 ppt group presented a significantly higher wet weight than the 39 ppt group ($p < 0.05$), and this was maintained until the end of the experiment ($400.2 \pm 8.6$ g at 12 ppt, $300.3 \pm 2.6$ g at 39 ppt). Length increased accordingly, and differences were statistically different from day 10 onwards (Figure 1B). The biometric results recorded at day 45 (trial end) are shown in Table 1, where the specific growth rate (SGR), daily growth rate (DGR), and weight gain (WG) were significantly higher in the 12 ppt group ($p < 0.05$), while the Fulton´s condition factor (K) showed no differences between groups. Thus, the SGRs at 12 and 39 ppt were $3.83 \pm 0.04$ and $3.30 \pm 0.02$, respectively.

**Table 1.** Growth-related parameters in *Argyrosomus regius* fingerlings acclimated to 12 and 39 ppt salinity conditions for 45 days. SGR: Specific growth rate; DGR: Daily growth rate; WG: Weight gain; K: Fulton´s condition factor. Data are expressed as the mean $\pm$ SEM (n = 120). Asterisks (*) indicate significant differences among groups ($p < 0.05$, Student's *t*-test).

| Salinity | SGR (% day$^{-1}$) | DGR (g day$^{-1}$) | WG (%) | K |
|---|---|---|---|---|
| **12 ppt** | $3.83 \pm 0.04$ | $0.41 \pm 0.01$ | $400.2 \pm 8.6$ | $1.20 \pm 0.05$ |
| **39 ppt** | $3.30 \pm 0.02$ * | $0.31 \pm 0.00$ * | $300.3 \pm 2.6$ * | $1.18 \pm 0.05$ |

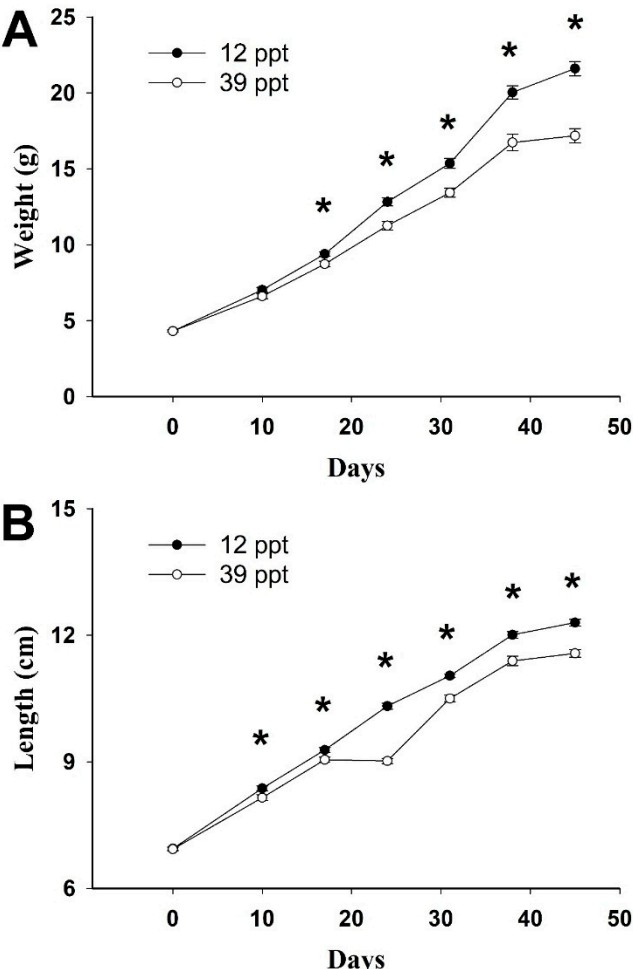

**Figure 1.** Body weight (**A**) and length (**B**) development in *Argyrosomus regius* fingerlings acclimated to 12 (black dots) and 39 (white dots) ppt salinity conditions for 45 days. Data are expressed as the mean ± SEM (*n* = 120). Asterisks (*) indicate significant differences among groups (*p* < 0.05, Student's *t*-test).

Plasma osmolality did not vary between 12 ppt (271 ± 2 mOsm kg$^{-1}$) and 39 ppt (274 ± 3 mOsm kg$^{-1}$). A range of different salinities from 5 to 39 ppt were prepared with the seawater and the freshwater employed in this study, and the results were in accordance with previous studies performed by our research group [17,30,35,36] with 161, 343, and 1159 mOsm kg$^{-1}$ for the salinities of 5, 12, and 39 ppt. As the water osmolality increased with environmental salinity (within the range from 5 to 39 ppt) fitting in a straight regression line ($r^2$ = 0.994, *p* < 0.05), and no variations were found in plasma osmolality between fish acclimated to both experimental salinities (12 and 39 ppt), we substituted the average plasma osmolality calculated for fingerlings (273 ± 2 mOsm kg$^{-1}$) in the regression line calculated for water osmolality, thus resulting in an iso-osmotic salinity for *A. regius* fingerlings of 9.2 ppt.

Branchial and renal Na$^+$/K$^+$-ATPase activities did not vary between both groups (*p* > 0.05, Student´s *t*-test). Specifically, Na$^+$/K$^+$-ATPase activity in the gills ranged between 11.7 ± 0.7 and 11.4 ± 0.9 µmol ADP mg prot$^{-1}$ h$^{-1}$ for the 12 and 39 ppt groups, respectively. In kidneys, Na$^+$/K$^+$-ATPase activity levels were 12.0 ± 0.7 and 13.0 ± 0.6 µmol ADP mg prot$^{-1}$ h$^{-1}$ for the 12 and 39 ppt groups, respectively.

At day 45, plasma cortisol concentrations (Figure 2) were significantly lower in the 12 ppt acclimated fingerlings as compared to the 39 ppt group (8.9 ± 0.9 and 12.9 ± 0.9 ng mL$^{-1}$, respectively; *p* < 0.05).

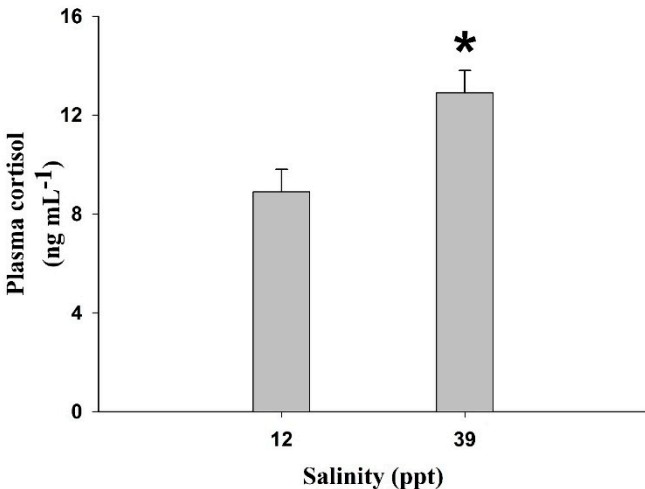

**Figure 2.** Plasma cortisol concentrations in *A. regius* fingerlings acclimated to 12 and 39 ppt salinity conditions for 45 days. Data are expressed as the mean $\pm$ SEM ($n$ = 9). Asterisks (*) indicate significant differences among groups ($p < 0.05$, Student's *t*-test).

The assessed plasma metabolites are shown in Table 2. The plasma concentrations of glucose, lactate, triglycerides (TAG), and free fatty acids (FFA) significantly decreased in the 12 ppt group relative to the 39 ppt group, but plasma proteins and free amino acids did not differ statistically. In this sense, plasma glucose levels at 12 and 39 ppt were 5.6 $\pm$ 0.5 and 15.5 $\pm$ 0.8 mM, respectively. Plasma TAG at 12 ppt (8.6 $\pm$ 0.6 mM) were half of the concentration at 39 ppt (16.9 $\pm$ 1.6 mM).

**Table 2.** Plasma metabolites in *A. regius* fingerlings acclimated to 12 and 39 ppt salinity conditions for 45 days. Data are expressed as the mean $\pm$ SEM ($n$ = 12). Asterisks (*) indicate significant differences among groups ($p < 0.05$, Student's *t*-test). TAG means triglycerides, FFA means free fatty acids.

| Salinity | Glucose (mM) | Lactate (mM) | TAG (mM) | FFA (mM) | Proteins (g dL$^{-1}$) | Amino Acids (mM) |
|---|---|---|---|---|---|---|
| **12 ppt** | 5.6 $\pm$ 0.5 | 1.9 $\pm$ 0.2 | 8.6 $\pm$ 0.6 | 9.6 $\pm$ 0.5 | 32.3 $\pm$ 1.3 | 4.2 $\pm$ 0.3 |
| **39 ppt** | 15.5 $\pm$ 0.8 * | 2.3 $\pm$ 0.2 * | 16.9 $\pm$ 1.6 * | 13.6 $\pm$ 1.5 * | 30.0 $\pm$ 1.1 | 4.2 $\pm$ 0.3 |

The hepatosomatic index (HSI) is presented in Table 3. The fingerlings acclimated to 39 ppt presented lower HSI values compared to 12 ppt fingerlings (5.8 $\pm$ 0.6% and 3.6 $\pm$ 0.1% for fish at 12 and 39 ppt, respectively; $p < 0.05$). The levels of liver-stored glycogen, glucose, and free amino acids are presented in Table 3. In the liver, there were significantly increased levels of all metabolites in the 12 ppt group compared to the 39 ppt group ($p < 0.05$).

**Table 3.** Hepatic parameters (hepatosomatic index (HSI) and metabolites) in *A. regius* fingerlings acclimated to 12 and 39 ppt salinity conditions for 45 days. Data are expressed as the mean $\pm$ SEM ($n$ = 12). Asterisks (*) indicate significant differences among groups ($p < 0.05$, Student's *t*-test).

| Salinity | HSI (%) | Glycogen (mg Liver$^{-1}$) | Glucose (mg Liver$^{-1}$) | TAG (mg Liver$^{-1}$) | Amino Acids (μmol Liver$^{-1}$) |
|---|---|---|---|---|---|
| **12 ppt** | 5.6 $\pm$ 0.6 | 32.7 $\pm$ 2.3 | 15.4 $\pm$ 1.3 | 5.5 $\pm$ 1.0 | 18.2 $\pm$ 1.7 |
| **39 ppt** | 3.6 $\pm$ 0.1 * | 22.2 $\pm$ 1.4 * | 9.8 $\pm$ 0.6 * | 2.0 $\pm$ 0.4 * | 12.0 $\pm$ 0.9 * |

## 3. Discussion

The present study is the first to propose that brackish water may be more appropriate for the culture of *A. regius* fingerlings than seawater. This information is relevant to the aquaculture industry

as optimizing environmental salinities for this species could improve growth rates, resulting in economic advantages.

Estuaries are associated with the life history of the meagre, and meagre juveniles are mostly found in brackish waters [10]. Moreover, responses to salinity are also influenced by the age of the fish [24]. These prior descriptions are supported by the present study, which found that three-month-old fingerlings presented better growth rates in brackish water (iso-osmotic salinities, 12 ppt) compared to seawater (SW, 39 ppt).

The length and weight of the A. regius fingerling groups differed from days 10 and 17 onwards of the salinity challenge, respectively, with higher growth rates at 12 ppt from these dates onwards. In the wild, A. regius, A. hololepidotus and A. japonicus fingerling are mostly associated with brackish waters ($\approx$ 12 ppt) [11,21,37], which is in line with the better growth performance observed in the present study at this environmental salinity. However, major differences were found when comparing A. regius to the related A. japonicus, a species that presents optimum growth at 35 ppt for individuals weighing less than 10 g [22]. This difference between closely-related fish species highlights the importance of separately studying each species to determine the best environmental salinity for growth [24].

The osmoregulatory data obtained by the present study provides support for the euryhalinity of *A. regius*, which was found to be able to be maintained within a wide range of environmental salinities. Specifically, fingerlings kept plasma osmolality values within a constant range (271–274 mOsm kg$^{-1}$) at 12 and 39 ppt, in good agreement with *A. inodorus* juveniles maintaining their plasma osmolality constant amongst the range from 15 to 35 ppt salinity [38]. However, *A. regius* juveniles showed higher plasma osmolality values, 387–400 mOsm kg$^{-1}$ when cultured in full-strength seawater [39]. Therefore, the better growth rates of fingerlings in iso-osmotic water could be associated with their lower osmolality of body fluids compared to juveniles. The reason for these differences in plasma osmolality between fingerlings and juveniles is still unknown, but it may be related to an evolutionary adaptation to favor the lower predator densities of lower salinity environments, or higher productivity rates in estuarine areas [40], or related to possible ontogenetic differences in ion-pump activities.

The branchial and renal Na$^+$/K$^+$-ATPase (NKA) enzyme, an important modulator of osmoregulatory processes that is responsible for major energy expenditures during osmoregulation [15,31,36,41], exhibited no changes between 12 and 39 ppt salinity. This result was similar to that previously described for other teleost species, such as *Sparus aurata* [41], and reflects that other osmoregulatory tissues such as the intestine have important roles in the maintenance of body fluids, as described before in *Solea senegalensis* and *Galaxias maculatus* [36,42]. Thus, ionocyte cells in osmoregulatory tissues may show differentiated ion-transport mechanisms rather than the NKA enzyme, highlighting the importance of further studies on this topic to fully elucidate the relative importance of osmoregulation on body growth. Moreover, this study did not measure drinking rates, which is an energetically costly mechanism that involves digestive tract desalinization to maintain plasma osmolality at high environmental salinities, and includes changes in the population of intestinal cells [36]. In light of the results from this study, it is not clear that osmoregulatory processes at 39 ppt salinity consume more energy that at 12 ppt, though fish growth is faster at the latter salinity. More studies are necessary to fully characterize the energy consumed by osmoregulatory tissues in meagre fingerlings acclimated at different environmental salinities.

The maintenance of proper homeostatic levels is achieved in fish through allostatic changes [43], and variations in environmental salinity are compensated through the expenditure of metabolic energy [33,44]. In this sense, the present study found that *A. regius* fingerlings acclimated to a salinity of 39 ppt mobilized and/or did not accumulate great amounts of metabolites as compared to fingerlings maintained in iso-osmotic environments (12 ppt). Cortisol acts as a glucocorticoid hormone in teleosts [32], so higher levels of this steroid in fingerlings acclimated to a salinity close to that in the Mediterranean Sea in summer (39 ppt) compared to those acclimated to an estuarine-like salinity (12 ppt) may indicate a greater consumption of energy metabolites. This hormone mobilizes energy metabolites, thus increasing carbohydrate consumption and lipid mobilization from the liver, the most important source of lipid deposits in *A. regius* [45]. Consequently, 39 ppt fingerlings demonstrated

decreased HSI values and lower growth performance compared to fingerlings at 12 ppt. Therefore, energy metabolites such as liver glycogen may be mobilized through the plasma to satisfy increased osmoregulatory demands rather than being stored, as also recorded for this species under natural environmental conditions [26].

## 4. Materials and Methods

### 4.1. Animals

*A. regius* fingerlings were provided by the El Toruño Center for Investigation and Formation in Fishery and Aquaculture (IFAPA) (El Puerto de Santa María, Cádiz, Spain). All experimental procedures complied with the guidelines of the University of Cadiz (Spain) and the European Union (86/609/EU) for the use of animals in research, and the Commission of Ethics and Animal Research of the University of Cadiz approved the experiment.

### 4.2. Experimental Design

All fingerling experiments were carried out in summer 2008 at the El Toruño IFAPA Center (El Puerto de Santa María, Cádiz, Spain). SW-acclimated *A. regius* fingerlings (94 days post-hatching, DPH; $4.32 \pm 0.07$ g body weight, mean $\pm$ SEM, $n = 240$) were randomly divided into eight tanks (125 L each, fingerling density of 1 g $L^{-1}$). Four tanks contained SW (39 ppt, $n = 120$), and four contained brackish water presumably close to the iso-osmotic point for this species (12 ppt, $n = 120$). Each tank included a closed-loop recirculation system with filtration (including physical, biological, and sand filters, together with an ultraviolet lamp before water was returned into the tanks), with roughly 20% of the tank volume changed daily. Brackish water (12 ppt) was obtained by mixing SW with dechlorinated tap water in 900-L tanks. During the experiment, fingerlings were maintained under natural photoperiod conditions (June–August; latitude 36°34′43″N) and a constant temperature ($22.4 \pm 0.1$ °C). Temperature, salinity, pH ($7.82 \pm 0.03$ at 39 ppt and $7.87 \pm 0.03$ at 12 ppt), and oxygen ($6.45 \pm 0.09$ mg $L^{-1}$ at 39 ppt and $7.58 \pm 0.08$ mg $L^{-1}$ at 12 ppt) were monitored daily, while concentrations of nitrite ($0.05 \pm 0.01$ mg $L^{-1}$ at 39 ppt and $1.69 \pm 0.97$ mg $L^{-1}$ at 12 ppt), nitrate ($4.13 \pm 0.15$ mg $L^{-1}$ and $36.41 \pm 3.50$ mg $L^{-1}$ at 12 ppt), and ammonia ($0.02 \pm 0.01$ mg $L^{-1}$ at 39 ppt and $0.03 \pm 0.01$ mg $L^{-1}$ at 12 ppt) were monitored weekly. No major changes in these parameters were observed over the experimental period. Weekly measurements of length and weight were taken as described below. Fingerlings were fed with commercial dry pellets once a day in proportion to 4% of their body weight (Skretting, Stavanger, Norway; Gemma 1,8 and Skretting CV-2 corvina). After 45 days under experimental salinities, fingerlings were sampled and growth, feeding, and physiological parameters were measured. No mortalities were recorded during the experiment.

### 4.3. Growth Parameters

At days 0 (start of the experiment, 94 DPH), 10 (104 DPH), 17 (111 DPH), 24 (118 DPH), 31 (125 DPH), 38 (132 DPH), and 45 (139 DPH), fingerlings were captured, anaesthetized with a slight dose of clove oil (0.125 mL $L^{-1}$), their length and weight were measured, and the fish were returned to their respective tanks. No mortalities were observed during this process. The evaluated parameters were the specific growth rate (SGR), daily growth rate (DGR), weight gain (WG), and Fulton's condition factor (K). At the end of the experimental period (day 45), the hepatosomatic index (HSI) was also calculated. The above parameters were calculated as follows:

$$\text{SGR (\% day}^{-1}) = 100 \times (\text{ln Wf-ln Wi}) \times \text{T}^{-1}$$

$$\text{DGR (g day}^{-1}) = (\text{Wf-Wi}) \times \text{T}^{-1}$$

$$\text{WG (\%)} = 100 \times ((\text{Wf-Wi}) \times \text{Wi}^{-1})$$

$$K = 100 \times (Wt\ Lt^{-3})$$

$$HSI = 100 \times (Wl\ Wt^{-1})$$

where Wt = total wet body weight (g); Wf = final wet body weight (g); Wi = initial wet body weight (g); Wl = wet liver weight; Lt = total body length (cm); and T = time elapsed between each sampling point (days).

### 4.4. Blood and Tissue Sampling

Blood and tissue samples were taken from fingerlings at day 45 post-transfer. Animals were fasted for 24 h before sampling. Fingerlings were deeply anaesthetized with clove oil (2 mL L$^{-1}$, 0.2% volume/volume), and then weighed and sampled. Blood was collected after caudal removal with heparinized micro-capillary tubes. Plasma for each fish was separated from whole blood by centrifugation (3 min, 4 °C at 10,000× *g*), immediately frozen in liquid nitrogen, and stored at −80 °C until analysis. From each fish, the second gill arch on the left side was excised, dried with absorbent paper, and a small portion was cut using fine-point scissors. A small portion of caudal kidney was also collected. These small portions were placed in 100 μL of ice-cold sucrose-ethylenediaminetetraacetic acid (EDTA)-imidazole buffer (150 mM sucrose, 10 mM EDTA, and 50 mM imidazole, pH 7.3) and frozen at −80 °C until analyses of Na$^+$/K$^+$-ATPase activity. The liver was excised, weighed, and snap-frozen at −80 °C. All procedures lasted less than 3 min per tank in order to ensure that the acute-stress response due to handling did not affect further analysis [46].

### 4.5. Plasma Measurements

Plasma and water osmolality were measured with a Fiske One-Ten vapor pressure osmometer (Fiske Associates, Advanced Instruments, Norwood, MA, USA). The iso-osmotic point was estimated according to previous studies [47] as the intersect of the iso-osmotic line and the regression lines between plasma and water osmolality. Plasma glucose, lactate, and TAG levels were measured on 96-well microplates using commercial kits (Spinreact, St. Esteve de Bas, Spain; Glucose-HK Ref. 1001200; Lactate Ref. 1001330; TAG Ref. 100131101). Total plasma proteins were determined using the Bicinchoninic Acid Protein Assay Kit #23225 (Pierce, Rockford, IL, USA) with bovine serum albumin as the standard. Total α-amino acid levels were assessed through colorimetric analysis using the ninhydrin method [48] adapted to microplates and using L-alanine as the standard. Free fatty acids were analyzed with a commercial kit (Wako Chemicals GmbH, Neuss, Germany) using oleic acid as the standard. All assays were carried out on a microplate reader (Bio-Tek Instruments, Winooski, VT, USA) using the KCjunior Data Analysis Software for Microsoft Windows XP. Plasma cortisol levels were measured using an enzyme-linked immunosorbent assay performed in microtiter plates (MaxiSorp, Nunc, Roskilde, Denmark), as previously described for other teleost species [49]. Validations showed that serial dilutions of untreated plasma diluted with distilled water displaced cortisol from the antibody in parallel with dilutions of the standards (results not shown).

### 4.6. Gill and Kidney Na$^+$/K$^+$-ATPase Activity

Gill and kidney Na$^+$/K$^+$-ATPase activity was determined as previously described [50], with modifications [51]. Briefly, incubation conditions were pH 7.5, 25 °C, 47.3 mM Na$^+$, 21.1 mM K$^+$, 2.63 mM Mg$^{2+}$, 0.5 mM ATP, 2.1 mM phosphoenol pyruvate, 0.5 mM NADH, 4.0 U lactate dehydrogenase mL$^{-1}$, 4.0 U pyruvate kinase mL$^{-1}$, 7.5 mM sucrose, and 0.5 mM EDTA. Na$^+$/K$^+$-ATPase activity was specifically inhibited by 0.5 mM ouabain (O-3125, Sigma-Aldrich, St. Louis, MO, USA). Incubation lasted 3–5 min, and the maximum slope of NADH degradation (measured at 340 nm wavelength) was assumed to be the maximum velocity of Na$^+$/K$^+$-ATPase activity.

*4.7. Liver Metabolite Levels*

Each excised and frozen liver was finely minced on an ice-cold Petri dish, placed in 5-mL polypropylene tubes, and homogenized by ultrasonic disruption with a MISONIC XL Sonicator (QSONICA L.L.C., Newtown, CT, USA) in 7.5 (*w/v*) volumes of ice-cooled 0.6 N perchloric acid, after which the homogenate was neutralized using the same volume of 1 M $KHCO_3$. The homogenate was centrifuged (30 min, $13,000 \times g$, 4 °C), and the supernatants were stored in different aliquots at $-80$ °C until use in the metabolite assays. Tissue glycogen concentrations were assessed using the method described before [52]. The levels of glucose after glycogen breakdown (i.e., after subtracting free glucose levels) were determined with a commercial kit (Spinreact, St. Esteve de Bas, Spain). Total $\alpha$-amino acid and TAG levels were determined as described above for the plasma samples. All parameters were referred to the sampled piece of liver employed for the analysis and to the total weight of each liver. Thus, the units shown were displayed as total amount of energy metabolites assessed per liver.

*4.8. Statistics*

Results were expressed as means $\pm$ standard error of the mean (SEM). Normality and homoscedasticity were analyzed using the Kolmogorov–Smirnov and the Levene tests, respectively. Two-way nested analysis of variance (ANOVA) was performed to evaluate inter-tank variability of quadruplicates for all parameters. Since no significant variability was determined due to quadruplicates in any of the dependent variables ($p \geq 0.25$), all four tanks were subsequently treated as a single group, and individual fish were considered as samples. Thus, differences due to environmental salinity were evaluated through a Student´s *t*-test in order to facilitate the comprehension of the results. Fingerling samples were pooled (n = 3 individuals per tank, n = 12 per treatment) to ascertain statistical differences in plasma variables, liver metabolites, and $Na^+/K^+$-ATPase activity. Analyses were conducted using Statistica v.7 software. Statistical significance was accepted at $p < 0.05$.

**5. Conclusions**

The present study indicates that knowledge on osmoregulatory requirements of *A. regius* at each developmental stage is crucial for the selection of the best environmental salinity for optimal growth in the aquaculture industry. This is especially relevant for fingerlings (5–20 g), which show better growth performance at an iso-osmotic salinity (12 ppt) compared to 39 ppt, the value normally observed in earth pond and Mediterranean Sea-cage rearing systems during summer. Future studies are encouraged to further improve the knowledge of the osmotic necessities of *A. regius* over its lifecycle, specifically those studies focused on the long-term effects of salinity on growth.

**Author Contributions:** Study conception and design: S.C. and J.M.M., I.R-J., A.B.T., L.V-C., J.A.M-S. and A.R-R. conducted the experiments. Acquisition of data: I.R-J., A.B.T., L.V-C. and J.A.M-S. Analysis and interpretation of data: I.R-J., A.B.T., L.V-C. and J.A.M-S. Drafting of manuscript: I.R-J. and A.B.T. Critial revision: I.R-J., A.B.T., L.V-C., J.A.M-S., A.R-R., S.C. and J.M.M.

**Funding:** This research was partially funded by grant AGL2013-48835-C2-1-R (Ministerio de Economía y Competitividad, Spain) awarded to J.M.M.

**Acknowledgments:** The authors wish to thank the El Toruño IFAPA Center (El Puerto de Santa María, Cádiz, Spain) for providing experimental fish.

**Conflicts of Interest:** The authors declare no competing or financial interests. The funding sponsors had no role in the design of the study, in the collection, analyses, or interpretation of data; in the writing of the manuscript, and in the decision to publish the results.

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
