# Peer review of "Environmental Salinity Affects Growth and Metabolism in Fingerling Meagre (Argyrosomus Regius)"

_fishes, doi:10.3390/fishes4010006_

Round 1
Reviewer 1 Report
In this manuscript, the authors present on the effects of salinity on growth, osmoregulation and metabolism in meagre fingerlings. Groups of fish were housed in replicate tanks and subjected to either 12 or 39 ppt for 45 days. Weight and length were measured near-weekly; remaining end points were measured after euthanasia at Day 45. The design is mostly straight-forward but a few points are questioned. Results are clearly reported, but were (in my opinion) in part unexpected and these unexpected outcomes could do with some additional discussion. English language is good, but a notable number of small grammatical errors exist (too numerous to list), so scrutiny by a native speaker is needed.
Title
In my opinion, this paper does not focus on stress – the measure of cortisol was done in an osmoregulation context and although exposure to higher salinity could well be stressful (“chronic stress”), the paper is not focusing on stress. I think the words ‘stress pathways’ need to be omitted from the title (aside from the word ‘stress’ and my perceived irrelevance to the study design, I do not know what a ‘stress pathway’ is – better to leave that confusion out altogether).
L22 and elsewhere: “fingerlings” and “juveniles” is used as adjective – in that case, please use the singular (fingerling growth, juvenile performance, etc).
L28: the last line of the Abstract would benefit from being a concluding statement – the statement “This study will contribute” makes for a weak finish, I believe.
L83:”…. due to energy savings in osmoregulatory processes”. Energy expenditure is not measured in this study, and therefore, this aim cannot be addressed. It should be removed from the objectives. It can be introduced in the discussion only as a possible explanation. The rationale for the measurement of the end points in this study needs to be weaved into the objectives/last para of Introduction.
Results
Throughout: only the statistical outcomes are presented (higher, lower, greater, smaller), but key findings (data) are not stated. It would be good to see some actual values included in the text description.
Fig 1: the analysis of length and weight data needs some further consideration – the data are repeated measurements, and therefore, differences in weight on Day X can be expected to result in differences in weight on Day Y. Maybe it is better to simply compare the slopes of the growth curves to avoid this problem?
Fig 2: authors report on cortisol levels, which is great. However, in the M&M, it is not clear what measures were taken to ensure that the acute stress response associated with capture and euthanasia did not confound these ‘chronic’ cortisol levels. This issue needs to be addressed in the paper.
Fig 1 & 2: I am concerned about the repeated, weekly sampling of the same fish – this is a repeated stressor that is likely to have a notable impact on the dataset as a confounding factor. It is not clear why this (in my view, questionable) approach was chosen. This, and the above point (fig 2) are my key concerns for this ms.
L161: can be avoided (repeats earlier detail)
L163: “differed FROM days 10 and 17 onwards….”?
L179-180: the differences in plasma omolality between juveniles and fingerlings at full SW are unexpected (it should be easier for large fish to osmoregulate?) and this should be discussed further, I believe, alongside possible ontogenetic differences in sodium pmp activities (if such data exist).
L189: no differences in sodium pump activity between fish in both salinities. It seems impossible to reconcile these findings with the statement (L73) that the sodium pump is the most energetically expensive part of the gill osmoregulatory machinery. Indeed, the authors (L83) are interested to explain growth performance in light of osmoregulatory energy, but then do not take this conflicting finding (no difference in ATPase activity, but notable differences in plasma metabolites and growth performance) into account; would not increased solute movement in the gut (to prevent dehydration) be matched with increased Na+, Cl- excretion in the gill? This also does not tie in with the statement on L197. I think that the discussion of these key, interacting factors would benefit from some notable revision.
L208-210: this concluding statement should be phrased as ‘suggestive of’, as consumption of energy sources (glycogen, amino acids, etc) was not measured in this study.
Author Response
Title
In my opinion, this paper does not focus on stress – the measure of cortisol was done in an osmoregulation context and although exposure to higher salinity could well be stressful (“chronic stress”), the paper is not focusing on stress. I think the words ‘stress pathways’ need to be omitted from the title (aside from the word ‘stress’ and my perceived irrelevance to the study design, I do not know what a ‘stress pathway’ is – better to leave that confusion out altogether).
Answer: The Title has been modified accordingly.
L22 and elsewhere: “fingerlings” and “juveniles” is used as adjective – in that case, please use the singular (fingerling growth, juvenile performance, etc).
Answer: Changes were done along the text as suggested.
L28: the last line of the Abstract would benefit from being a concluding statement – the statement “This study will contribute” makes for a weak finish, I believe.
Answer: The last sentence of the Abstract was modified accordingly.
L83:”…. due to energy savings in osmoregulatory processes”. Energy expenditure is not measured in this study, and therefore, this aim cannot be addressed. It should be removed from the objectives. It can be introduced in the discussion only as a possible explanation. The rationale for the measurement of the end points in this study needs to be weaved into the objectives/last para of Introduction.
Answer: The aim of the study was modified, as suggested.
Results
Throughout: only the statistical outcomes are presented (higher, lower, greater, smaller), but key findings (data) are not stated. It would be good to see some actual values included in the text description.
Answer: Values of some parameters were included in the text.
Fig 1: the analysis of length and weight data needs some further consideration – the data are repeated measurements, and therefore, differences in weight on Day X can be expected to result in differences in weight on Day Y. Maybe it is better to simply compare the slopes of the growth curves to avoid this problem?
Answer: The Statistics of length and weight did not include repeated measurements. Two-way nested analysis of variance (ANOVA) was performed to evaluate inter-tank variability of quadruplicates for all parameters. Since no significant variability was determined due to quadruplicates in any of the dependent variables (p ≥ 0.25), all four tanks were subsequently treated as a single group, and individual fish were considered as samples. Thus, differences due to environmental salinity were evaluated through a Student´s t-test in order to facilitate comprehension of the results.
Thus, the analysis at each sampling time was done as a Sudent´s t-test between both groups.
Fig 2: authors report on cortisol levels, which is great. However, in the M&M, it is not clear what measures were taken to ensure that the acute stress response associated with capture and euthanasia did not confound these ‘chronic’ cortisol levels. This issue needs to be addressed in the paper.
Answer: This issue is now addressed in the M&M and a reference supporting our methodology has been also included.
Fig 1 & 2: I am concerned about the repeated, weekly sampling of the same fish – this is a repeated stressor that is likely to have a notable impact on the dataset as a confounding factor. It is not clear why this (in my view, questionable) approach was chosen. This, and the above point (fig 2) are my key concerns for this ms.
Answer: Fish were weekly sampled in order to optimize food ration. Meagre of this size shows high growth rates that may be constantly (weekly) controlled in order to ensure that daily 4% of their body weight was offered as food. The experiments were conducted with the best zootechnical approaches available in that moment. As welfare of fish is actually a matter of concern, fish growth could nowadays be controlled remotely, through video analysis, but that technology was not available to us during the experiments. However, fish from our study ate properly one hour after sampling, showing a good recovery from the possible stress due to handling.
L161: can be avoided (repeats earlier detail)
Answer: We do not get what the Reviewer suggests. The notion that lower salinity environments may favour lower predator densities was not described earlier in the text. May we please ask the Reviewer what does she/he suggest?
L163: “differed FROM days 10 and 17 onwards….”?
Answer: The sentence was modified according to the suggestion.
L179-180: the differences in plasma omolality between juveniles and fingerlings at full SW are unexpected (it should be easier for large fish to osmoregulate?) and this should be discussed further, I believe, alongside possible ontogenetic differences in sodium pmp activities (if such data exist).
Answer: We were also wondering why juveniles and fingerlings of the same species show such differences in plasma osmolality. The suggestion done by the Reviewer results interesting and deserves further studies. However, though the aim of this study was to establish if environmental salinities close to the iso-osmotic point for this species results in better growth rates, we considered far from it to discuss ontogenetic differences in ion-pumps activities. We have included a sentence in the Discussion section regarding the issue suggested by the Reviewer, hoping that plasma osmolality in the same species at different life stages results of interest for future studies.
L189: no differences in sodium pump activity between fish in both salinities. It seems impossible to reconcile these findings with the statement (L73) that the sodium pump is the most energetically expensive part of the gill osmoregulatory machinery. Indeed, the authors (L83) are interested to explain growth performance in light of osmoregulatory energy, but then do not take this conflicting finding (no difference in ATPase activity, but notable differences in plasma metabolites and growth performance) into account; would not increased solute movement in the gut (to prevent dehydration) be matched with increased Na+, Cl- excretion in the gill? This also does not tie in with the statement on L197. I think that the discussion of these key, interacting factors would benefit from some notable revision.
Answer: The Introduction and Discussion sections were modified to accommodate for the suggestion.
L208-210: this concluding statement should be phrased as ‘suggestive of’, as consumption of energy sources (glycogen, amino acids, etc) was not measured in this study.
Answer: The suggestion was accepted and the sentence modified accordingly.
Reviewer 2 Report
A well planned and made study. It is worth of publication.
Author Response
We want to thank the Reviewer for her/his comment.
Reviewer 3 Report
This study aims to evaluate the ability of early stages of meagre to grow at salinities close to its isosmotic point, in contrast to seawater where most of the rearing actually occurs. It is a good fit for the volume, but there are issues that need clarification before acceptance for publication.
It has been pointed out many times that living at an isosmotic environment should reduce the costs of osmoregulation and consequently energy could be reallocated to other functions, storage, or growth. Interestingly not all studies have shown this, and the reasons are diverse.
In this case, the data shows that living at a reduced salinity seems to be advantageous for fish growth. The experimental design appears to be well implemented and the analytical methods are according the current practices in these types of study. The introduction clearly places the study in the state of the art, although the data is from 2008, since not many other developments have occurred in relation to this issue in meagre culture. However, some sentences need better contextualization, as they are left hanging. Please see specific comments below.
The discussion touches the relevant topics raised by the data and shows good use of the available literature. Nonetheless I feel some of the results need to be better explained. The lack of a difference in plasma osmolality when environments differ in about 1200 mOsmol may indicate these fish are very good osmoregulators (although in many studies, fish such as O, mossambicus or F. heteroclitus, which are completely euryhaline, still denote some small differences in osmolality). However the roles of the branchial or renal NKA, which do not differ among salinities, are then puzzling. How can the environmental pressure to gain or loose ions and water be accommodated without alteration in these crucial mechanisms. The authors indicate that intestine may be the critical organ, but there is no evidence in this study. Would this entail important changes in drinking rates, energetically costly esophageal desalinization to maintain such low osmolality? Where would be Na and Cl extruded if not the gills? Although there are changes in some plasma parameters, these do not seem to amount to a difference great enough to alter osmolality, and still they are higher in high salinity plasma. I would like to see these issues better exploited as they are at the core of the study – with the data shown it is clear that energy is being used differently among the two groups but it is not clear that osmoregulatory processes are using that energy.
Specific comments:
Line 39. All productions are restricted to some countries - please specify the group, conditions and/or why… not adequate conditions, not adequate market value… as it is this is not informative.
Line 54 and others. Juvenile. In this context there is no plural. In line 58 juveniles is correct
Line 62… that the salinity for optimum growth can vary
Line 67. This paragraph is overall correct although generalist. Not all species behave eactly this way, and it should be pointed out that after change the homeostatic point it is probably not the same as before, with the consequent alteration to the activity of the mechanisms involved. How does this paragraph fits with the results? “Gills (and to a minor extent, kidney and digestive tract) are the most energetically demanding tissue in terms of osmoregulation”
Line 163. Not only at 10 and 17 but starting on… rephrase
Line 179. Juvenile
Line 180. When would this transition from a “brackish water to sea water fish” occur? Could this be considered some sort of “meagre smoltification” with important changes in hormonal regulation?
Line 181. how would lower plasma osmolality be related to increased growth ? this is reaching… please provide explanation or references that can substantiate this.
Line 184. to favor the permanence of more susceptible stages in an environment with lower predatory pressure, such as the low salinity waters... (is this true? Less predators in estuaries?)
Line 191. Contextualize reference 41 in this topic. How would be salts eliminated?
Line 201-204. Why not surprising (is high salinity more stressful or energy demanding in general? and how does it fit without the lack of changes in NKA?
Line 291-293- formatting
Line 340-342. Are there any published growth rates/growth parameters for the fingerlings maintained in those production culture systems? How does it compare to the data shown here?
Line 499. Check reference 41. Are the more authors? et al? Please confirm remaining reference entries
Author Response
The discussion touches the relevant topics raised by the data and shows good use of the available literature. Nonetheless I feel some of the results need to be better explained. The lack of a difference in plasma osmolality when environments differ in about 1200 mOsmol may indicate these fish are very good osmoregulators (although in many studies, fish such as O, mossambicus or F. heteroclitus, which are completely euryhaline, still denote some small differences in osmolality).
Authors: The difference between the 12 ppt and the 39 ppt environments was of 816 mOsm kg-1. This great difference in environmental osmolality makes meagre a good osmoregulatory species, as its plasma osmolality does not show differences in fish acclimated at those salinities.
However the roles of the branchial or renal NKA, which do not differ among salinities, are then puzzling. How can the environmental pressure to gain or loose ions and water be accommodated without alteration in these crucial mechanisms. The authors indicate that intestine may be the critical organ, but there is no evidence in this study. Would this entail important changes in drinking rates, energetically costly esophageal desalinization to maintain such low osmolality?
Authors: The Discussion section was modified according to this suggestion.
Where would be Na and Cl extruded if not the gills?
Authors: Na and Cl are mostly extruded through the gills.
Although there are changes in some plasma parameters, these do not seem to amount to a difference great enough to alter osmolality, and still they are higher in high salinity plasma. I would like to see these issues better exploited as they are at the core of the study – with the data shown it is clear that energy is being used differently among the two groups but it is not clear that osmoregulatory processes are using that energy.
Authors: We have changed the aim of the study according to the suggestion of other Reviewer. Now, osmoregulation processes are not the main objective of this study. We have modified the text accordingly.
Specific comments:
Line 39. All productions are restricted to some countries - please specify the group, conditions and/or why… not adequate conditions, not adequate market value… as it is this is not informative.
Authors: The sentence has been modified for clarity.
Line 54 and others. Juvenile. In this context there is no plural. In line 58 juveniles is correct
Authors: We want to thank the Reviewer for his/her help with the English grammar. We have carefully revised all text long.
Line 62… that the salinity for optimum growth can vary
Authors: The sentence has been modified accordingly.
Line 67. This paragraph is overall correct although generalist. Not all species behave eactly this way, and it should be pointed out that after change the homeostatic point it is probably not the same as before, with the consequent alteration to the activity of the mechanisms involved. How does this paragraph fits with the results? “Gills (and to a minor extent, kidney and digestive tract) are the most energetically demanding tissue in terms of osmoregulation”
Authors: The paragraph has been modified.
Line 163. Not only at 10 and 17 but starting on… rephrase
Authors: The sentence was rephrased.
Line 179. Juvenile
Authors: Change accepted.
Line 180. When would this transition from a “brackish water to sea water fish” occur? Could this be considered some sort of “meagre smoltification” with important changes in hormonal regulation?
Authors: Juvenile meagre is found at brackish waters, while adults are found at sea water. The Reviewer is right while assuming important changes in hormonal regulation may be involved. Thus, the manuscript of Mohammed-Geba et al. (Molecular performance of Prl and Gh/Igf1 axis in the Mediterranean meager, Argyrosomus regius, acclimated to different rearing salinities. Fish Physiol Biochem 2017, 43, 203-216) reinforced this idea. Future studies may be directed towards the pituitary control of osmoregulation in this species.
Line 181. how would lower plasma osmolality be related to increased growth ? this is reaching… please provide explanation or references that can substantiate this.
Authors: To the best of our knowledge, osmoregulation in marine fish is a costly process that, itself, may be involved in somatic growth. Moreover, environmental salinity also induces changes in the endocrine control of growth. To what extend plasma osmolality influences growth in meagre is unknown to us? The paragraph in the new version of the manuscript was modified for clarity.
Line 184. to favor the permanence of more susceptible stages in an environment with lower predatory pressure, such as the low salinity waters... (is this true? Less predators in estuaries?)
Authors: The cited manuscript of Wada et al. (2007) said so. However, we have included that estuarine areas may be also better for fingerlings due to their higher productivity rates.
Line 191. Contextualize reference 41 in this topic. How would be salts eliminated?
Authors: The paragraph was modified for clarity.
Line 201-204. Why not surprising (is high salinity more stressful or energy demanding in general? and how does it fit without the lack of changes in NKA?
Authors: The sentence was modified for clarity.
Line 291-293- formatting
Authors: Done.
Line 340-342. Are there any published growth rates/growth parameters for the fingerlings maintained in those production culture systems? How does it compare to the data shown here?
Authors: We were unable to find out information regarding this issue.
Line 499. Check reference 41. Are the more authors? et al? Please confirm remaining reference entries
Authors: Thanks for the careful revision process. The “et al.” was written automatically by the formatting program “EndNote” as the number of authors of that paper was above 10. However, we have manually included the missing authors.